# Bone Fragility in Gastrointestinal Disorders

**DOI:** 10.3390/ijms23052713

**Published:** 2022-02-28

**Authors:** Daniela Merlotti, Christian Mingiano, Roberto Valenti, Guido Cavati, Marco Calabrese, Filippo Pirrotta, Simone Bianciardi, Alberto Palazzuoli, Luigi Gennari

**Affiliations:** 1Department of Medical Sciences, Azienda Ospedaliera Universitaria Senese, 53100 Siena, Italy; 2Department of Medicine Surgery and Neuroscience, University of Siena, 53100 Siena, Italy; christian.mingiano@gmail.com (C.M.); guido.cavati@gmail.com (G.C.); m.calabrese5@student.unisi.it (M.C.); pirrotta@student.unisi.it (F.P.); s.bianciardi@gmail.com (S.B.); 3Deparment of Surgery, Perioperative Medicine Unit, Azienda Ospedaliera Universitaria Senese, 53100 Siena, Italy; r.valenti@ao-siena.toscana.it; 4Cardiovascular Disease Unit, Division of Cardiology, Department of Medical Biotechnologies, Azienda Ospedaliera Universitaria Senese, 53100 Siena, Italy; alberto.palazzuoli@ao-siena.toscana.it

**Keywords:** inflammatory bowel disease, osteoporosis, celiac disease, *Helicobacter pylori*, chronic gastritis and peptic ulcer disease, gastric cancer and gastrectomy, dysbiosis

## Abstract

Osteoporosis is a common systemic disease of the skeleton, characterized by compromised bone mass and strength, consequently leading to an increased risk of fragility fractures. In women, the disease mainly occurs due to the menopausal fall in estrogen levels, leading to an imbalance between bone resorption and bone formation and, consequently, to bone loss and bone fragility. Moreover, osteoporosis may affect men and may occur as a sequela to different diseases or even to their treatments. Despite their wide prevalence in the general population, the skeletal implications of many gastrointestinal diseases have been poorly investigated and their potential contribution to bone fragility is often underestimated in clinical practice. However, proper functioning of the gastrointestinal system appears essential for the skeleton, allowing correct absorption of calcium, vitamins, or other nutrients relevant to bone, preserving the gastrointestinal barrier function, and maintaining an optimal endocrine-metabolic balance, so that it is very likely that most chronic diseases of the gastrointestinal tract, and even gastrointestinal dysbiosis, may have profound implications for bone health. In this manuscript, we provide an updated and critical revision of the role of major gastrointestinal disorders in the pathogenesis of osteoporosis and fragility fractures.

## 1. Introduction

Osteoporosis is a systemic skeletal disease affecting affect up to 50% of postmenopausal women and 20% of men older than 50 years [1,2]. The disorder is characterized by compromised bone mass and strength consequently, leading to an increased risk of fracture for even minimal trauma. The most frequent sites of fragility fractures are the vertebral bodies (especially at the level of the lumbar spine), the proximal femur (at the level of the neck or trochanter), and the distal end of the radius (“Colles” fractures); more rarely, fractures affect the humerus or other skeletal sites. Each year, more than 1.5 million people suffer hip, vertebral, and wrist fractures due to osteoporosis. These fractures have significant clinical consequences, both in terms of mortality and disability, with high health and economic burden worldwide [1,2].

Bone is a metabolically active tissue in which the processes of bone formation by the osteoblasts and osteoclast-induced bone resorption are continuous throughout life, thus ensuring that the bone tissue is continuously renewed and normal skeletal structure is maintained [3,4]. Moreover, in order to prevent bone damage, this remodeling cycle adapts bone structure, and hence bone strength, to its loading circumstances [4]. This action is coordinated by the osteocyte network that directs bone remodeling in relation to strain or microdamage [4,5]. Under normal conditions, mineralized old bone is removed and replaced by an equivalent amount of new bone matrix, which then undergoes mineralization. With aging, as well as in the presence of other conditions or disorders, an imbalance in bone remodeling may occur, leading to osteoporosis and bone fragility [4,6]. In women, osteoporosis and fractures mainly occur due to the menopausal fall in estrogen levels, leading to an imbalance between bone resorption and bone formation and, consequently, to net bone loss with each remodeling cycle. A decrease in estrogen production from androgen precursors has also been supposed to be a relevant mechanism of bone loss and osteoporosis in aging men [7,8,9,10]. Secondary osteoporosis may also occur in both genders and include drug-induced forms (i.e., glucocorticoids, immunosuppressive, and hormone deprivation therapies) or chronic disorders, such as endocrine diseases (hypogonadism, Cushing’s syndrome, hyperthyroidism, primary and secondary hyperparathyroidism (hyperPTH), acromegaly), autoimmune diseases, malabsorption syndromes, and neuro–muscular diseases.

In this review, we revise the role of gastrointestinal disorders in the pathogenesis of osteoporosis and fragility fractures.

## 2. Inflammatory Bowel Disease (IBD)

Inflammatory bowel disease (IBD) is a chronic intestinal disorder that is classified as one of two subtypes: Crohn’s disease and ulcerative colitis. Ulcerative colitis (UC) affects exclusively the colon, with superficial mucosal inflammation that extends proximally in a contiguous manner, and can lead to ulcerations, bleeding, toxic megacolon, and fulminant colitis. In contrast, Crohn’s disease (CD) can affect any part of the digestive tract, mainly in a noncontiguous manner, and is characterized by transmural inflammation, which can complicate in fibrotic strictures, fistulas, and abscesses [11]. IBD often begins during adolescence and young adulthood, with approximately 25% of patients presenting before the age of 20 years, [12] and is often characterized by periods of active disease or remission. Importantly, the global prevalence of IBD has increased substantially from 1990 to 2017, posing a heightened social and economic burden on governments and health systems [13].

The pathophysiology of IBD is not completely clear, but involves complex genetic, environmental, epithelial, microbial, and immune factors. In particular, the resulting bowel inflammation seems to occur due to the dysregulation of the immune system in response to changes in commensal (nonpathogenic) intestinal flora [14]. Genetic studies have shown that host–microbe interactions play a prominent role in the pathogenesis of both UC and CD, and involve genomic regions that regulate microbial defense and intestinal inflammation [15]. Clinical, endoscopic, histologic, and radiologic features are used to diagnose one form from another. The most common endoscopic test used for diagnosis is colonoscopy, which may differentiate the two different pathologies. UC is characterized by continuous and symmetric inflammation beginning in the rectum and extending proximally to varying degrees. There is often a gradation of disease that is most severe in the rectum and disappears proximally, ending in a sharp demarcation between affected and normal tissue. However, there are no definitive macroscopic alterations to diagnose UC. IBD that involves non-colon areas of the gastrointestinal tract is pathognomonic for CD. The macroscopic findings are characterized as patchy, asymmetric, focal, and discontinuous alterations of the mucosae [16]. The pharmacological management of IBD includes aminosalicylates, glucocorticoids, and immune-modulators, as well as targeted biologic therapies that act through the neutralization of cytokines that promote inflammation (e.g., anti-tumor necrosis factor (TNF) antibodies) or regulate the differentiation and function of specialized immune subsets (e.g., anti-IL-12 and anti-IL-23 antibodies), or induce the blockage of signal transduction cascades (e.g., Janus kinase (JAK) inhibitors), or the modulation of lymphocyte trafficking (e.g., anti-α4β7 integrin antibodies) [16].

### 2.1. Epidemiology of Osteoporosis and Fractures in IBD

Both UC and CD cause extra-intestinal manifestations (e.g., involving the eyes, skin, liver, bone, and joints), which can be found in 25% to 40% of patients [16]. Low bone mineral density (BMD), as assessed by dual X-ray absorptiometry, is considered among the general complications in patients with IBD (both those with CD and those with UC). According to the available literature, variable incidences of osteoporosis (BMD T-score below −2.5 SD) and osteopenia (BMD T-score between −1.0 and −2.5 SD) have been described in patients with IBD, ranging from 2% to 56%, with up to 40% estimated to have a higher risk of fracture than the general population [17,18,19]. In a recent meta-analysis, significantly higher relative risks of 1.38 (95% CIs, 1.11–1.73) for all fractures and 2.26 for vertebral fractures (95% CIs, 1.04–4.90) were reported in IBD patients than in the normal population [20]. In a subsequent meta-analysis, a similar increase in risk was described in IBD patients concerning vertebral fractures (OR = 2.21; 95% CI, 1.39–3.50), but not overall fractures (OR = 1.08; 95% CI, 0.72–1.62) [21]. Moreover, IBD patients with fractures were more commonly on steroids compared with those without fractures (OR = 1.47; 95% CI, 0.99–2.20). Some studies suggested a higher fracture risk in CD than in UC [22]. In addition, patients with childhood-onset IBD might be at the highest risk for low BMD and fractures [23]. In this respect, a recent study demonstrated that not only did young IBD cases show low BMD at all skeletal sites as compared to controls, but the risk of fracture was independently associated with trabecular bone micro-architectural parameters assessed by high-resolution peripheral quantitative computed tomography [24]. Thus, notwithstanding the differences in the various studies and the variable prevalence of osteoporosis (as defined by BMD measurement), patients with IBD tend to have a greater risk of fracture, especially at the vertebral level. This risk appears similar across genders and generally comparable between CD and UC.

### 2.2. Pathogenesis of Bone Fragility in IBD

The understanding of IBD-induced bone loss is far from complete, but definitely involves multiple mechanisms, either related or unrelated to the inflammatory condition (Figure 1). Appropriate chemically and genetically induced animal models have been employed in order to better define the characteristics of IBD-induced bone loss. Among the chemically induced models, both dextran sulfate sodium (DSS) and 2,4,6-trinitrobenzene sulfonic acid (TNBS) increase inflammatory disease severity and reduce the bone formation rate parameters, with an increase in bone loss. Moreover, reduced cortical thickness and deteriorated trabecular microstructure were observed in these IBD models [25]. In humans, numerous risk factors for the development of bone fragility in IBD have been suggested, such as genetic factors, malnutrition or impaired absorption of nutrients, calcium and vitamin D deficiency, sex hormone deficiency, a dysbiotic intestinal microbiome, and pharmacotherapy, particularly glucocorticoid therapy, which have to be considered in the management of such patients [26,27,28]. Among other risk factors for osteoporosis, it is well known that cigarette smoking induces oxidative stress, which may play an important role in IBD [29]. Likewise, nicotine and chemical compounds within tobacco smoke may impair bone quality, both directly and indirectly, modulating the receptor activator of the nuclear factor–kappa B (RANK)–receptor activator for the nuclear factor–kappa B ligand (RANKL)–osteoprotegerin (OPG) pathway (the master regulator of osteoclast activity and bone resorption), intestinal microbiota composition, and calcium-phosphate balance. Moreover, constant cigarette use interferes with the production of the protective mucus and inhibits the repair processes in the intestine, thus leading to impaired absorption of nutrients. Interestingly, smoking might affect patients with CD and UC in different ways; it seems to protect against UC, whereas, on the other hand, it increases the risk of CD development [29]. As already mentioned, the RANK–RANKL–OPG pathway is considered a key element of osteoporosis pathogenesis among patients suffering from IBD [30]. Generally, inflammation and increased pro-inflammatory cytokines (i.e., IL-6, IL-1, IL-17, and TNF-α) affect RANKL production in IBD patients, thereby promoting osteoclastogenesis and bone resorption [28,31]. This seems to be further heightened by the compounds of cigarette smoke, such as nicotine [31]. Additionally, smokers have a lower OPG level and higher RANKL/OPG ratio when compared with non-smokers [32], while nicotine disturbs the hydroxylation of vitamin D [33]. In contrast, osteoblast activity and bone formation were decreased in intact bone explants in the presence of serum derived from CD patients, even from those cases in disease remission [28]. Indeed, substantial inter individual variation, e.g., due to cytokine gene polymorphisms or a variable balance in the Th1–Th2 signature of the inflammatory response (with Th1 promoting and Th2 counteracting bone resorption), might finally impact global bone health in IBD patients [28]. This might also be dependent on the prostaglandin concentrations. In fact, prostaglandin E2, which is released in the body during inflammation, appears to protect epithelial cells from a special form of cell death in the intestinal barrier, named necroptosis, thus modulating the inflammatory burden in IBD. Patients with high levels of prostaglandin receptor EP4 on the epithelial cell surface showed a milder progression of intestinal inflammation than those with low levels of EP4, also by blocking the penetration of bacteria [34].

Patients with IBD, and especially those with a high disease burden, have various forms of malnutrition. In this respect, nutritional aspects, such as adequate calcium and vitamin D intake, seem to be crucial factors in increasing bone strength and reducing the risk of osteoporosis in this patient population [35]. A properly balanced diet, comprising an adequate supply of proteins, fats, carbohydrates, and macro- and micronutrients, might prevent bone loss and decrease the risk of fractures. Numerous studies indicate that sufficient calcium intake through the use of milk or dairy products may be helpful in patients with IBD, providing proper bone mineralization and increasing BMD [36,37,38]. Vitamin D deficiency is also common in IBD patients due to different mechanisms (e.g., intestinal malabsorption, decreased nutritional intake, and/or decreased sunlight exposure), and this negatively impacts the calcium balance and leads to secondary hyperPTH. Given the immune-modulatory role of vitamin D, this might also negatively impact the IBD disease course [39].

Malnutrition also leads to a low body mass index, which is commonly associated with osteopenia and decreased muscle strength, and might also contribute to sarcopenia, which is also commonly observed in patients suffering from IBD [40]. Sarcopenia is a syndrome associated with a reduction of skeletal muscle mass, leading to decreased muscle strength [41]. In the past, sarcopenia was thought to be a part of aging, specifically for older people. However, sarcopenia may also affect the young population. Its pathogenesis is multifactorial, and, in addition to aging and malnutrition, includes systemic inflammation, mitochondrial disorders, increase proteolysis, and insulin resistance [42]. The occurrence of sarcopenia has been strongly associated with osteopenia and osteoporosis in patients suffering from IBD. In fact, both sarcopenia and osteoporosis may be dependent on common mechanisms, such as low physical activity, oxidative stress, and chronic inflammation [43]. Moreover, sarcopenia constitutes a predictive factor for surgical intervention in IBD patients, being associated with an increased risk of severe postoperative complications, such as anemia, peri-operative sepsis, or venous thrombosis, and requiring more frequent hospitalizations after surgery [44].

Finally, patients with IBD are often treated with glucocorticoids, which have a further negative impact on bone health, increasing fracture risk and morbidity, particularly in elderly individuals. The greatest effect is usually seen in the initial months of treatment and with high dosages, especially in areas of trabecular bone, such as the vertebrae. This is mainly related to a decrease in bone formation due to the negative direct effects of glucocorticoids on osteoblasts and osteocytes [45,46]. However, the skeletal impact of low-dose glucocorticoid treatment is still controversial, with some studies showing an increased fracture risk with prednisone dosages as low as 2.5 mg/day [47]. Indeed, the real impact of glucocorticoid itself on the skeletal outcomes in IBD remains uncertain, at least in a subset of patients, given the potential benefits of treatment on the suppression of the inflammatory state. This might explain why some studies showed modest negative effects of high 10 mg/day prednisolone dosages on bone density and the vertebral fracture rates [48]. On the other hand, the use of anti-TNF-α agents has neutral or even positive effects on the maintenance of bone mass and the prevention of fracture in IBD due to its anti-inflammatory effects and possibly to its direct effects on bone metabolism [49,50,51,52,53]. The skeletal effects of other medications commonly used for IBD, such as mesalazines or newly developed immune-suppressants or biologics (e.g., vedolizumab), have not yet been investigated in detail.

### 2.3. Management of Bone Fragility in IBD

In order to improve bone health in IBD patients, dietary and lifestyle measures may be advocated and might be helpful for fracture prevention. These should mainly include physical activity and regular exercise, avoidance of smoking or excessive alcohol intake, and optimization of nutritional status. The latter should include calcium (500–1000 mg/day) and vitamin D (800 IU/day) supplementation, particularly in patients under chronic glucocorticoid treatment and/or low BMD. The optimization of nutritional status might be of particular relevance for bone health in pediatric and young IBD patients. Moreover, the identification of high-risk individuals is recommended in order to prevent osteoporotic fractured. In this respect, the American College of Gastroenterology suggested a cost-effective BMD screening in IBD patients with preexisting fragility fractures, women aged 65 or men aged 70 and older, and those patients with prior or current use of glucocorticoids or with other conventional risk factors for osteoporosis [54]. A slightly different approach was suggested by the American Gastroenterology Association that recommended a BMD analysis in postmenopausal women and men aged 50 years or older, patients with a history of vertebral fractures, chronic glucocorticoid use (3 months or more), or hypogonadism [55]. Conversely, the European Crohn and Colitis Organization did not suggest a different approach for IBD cases to that of the general population [56].

Bisphosphonates are among the drugs most often used in the treatment of osteoporosis or osteopenia and have been specifically investigated in some trials of patients with IBD. In a 2014 meta-analysis and systematic review of 19 controlled trials to evaluate the efficacy and safety of medical therapies for osteoporosis in patients with IBD, bisphosphonates were effective in increasing the BMD at the lumbar spine (standard difference in means, 0.51; 95% confidence interval, 0.29–0.72) and hip (standard difference in means, 0.26; 95% confidence interval, 0.04–0.49), as compared with the control treatment, with comparable tolerability, and the risk of vertebral fractures was also reduced (OR = 0.38, 95% CIs, 0.15–0.96) [57]. Subgroup analysis demonstrated that bisphosphonate therapy was significantly more effective than the control treatment in either short-term and long-term treatment groups, adult patients, patients with active or quiescent disease, studies targeting CD or both UC and CD, and for oral and intravenous treatment, respectively. There was insufficient evidence to assess the efficacy of calcium plus vitamin D, calcitonin, or other treatment regimens. Similar results were derived in a subsequent meta-analysis with a restricted number of randomized controlled trials (CTs; *n* = 11), showing a significant decrease in the incidence of overall fractures (OR = 0.30, 95% CI, 0.13–0.69) and vertebral fractures (OR = 0.38, 95% CIs, 0.16–0.93) with bisphosphonates compared to a placebo [58].

## 3. Celiac Disease

Celiac disease is a common autoimmune enteropathy triggered by dietary gluten and affecting up to 1% of the population worldwide, with two-fold higher prevalence in females [59,60,61]. This disorder is characterized by intestinal inflammation and villous atrophy upon exposure to gluten in genetically susceptible individuals and is also characterized by a variety of intestinal and extra-intestinal manifestations [59]. While some individuals may present with typical gastrointestinal symptoms (e.g., abdominal pain, abdominal distension, and diarrhea, together with weight loss and failure to grow in children), others may present without gastrointestinal complaints, but exhibit complications of celiac disease, such as anemia (due to iron deficiency), low BMD, elevated liver enzymes, leanness, prolonged fatigue, infertility, and, less frequently, unexplained neuropsychiatric disorder [60,61]. Importantly, the proportion of the overt, symptomatic, gastrointestinal cases has decreased over time, while there is an increasing number of patients presenting an extra-intestinal phenotype [59]. Moreover, celiac disease may also occur in the presence of other autoimmune disorders, such as type 1 diabetes, autoimmune thyroid disease, and dermatitis herpetiformis, as well as in patients with irritable bowel disease. Diagnosis is mainly based on specific serology (namely endomysial antibodies, transglutaminase antibodies, and deamidated antigliadin antibodies) and duodenal histology, both procedures combined can confirm the disease in the vast majority of cases [60].

### 3.1. Epidemiology of Bone Fragility in Celiac Disease

Skeletal involvement in celiac disease has been extensively studied and may include a variety of conditions spanning from secondary hyperPTH and osteomalacia to osteoporosis [61]. However, the prevalence of osteomalacia in celiac disease remains unknown.

The presence of low BMD levels at diagnosis, within the osteopenic and osteoporotic range, has been variably reported in 38–72% of cases [62]. Likewise, a 43–92% greater risk of fragility fractures has been described in patients with clinically diagnosed celiac disease compared with controls, albeit a large heterogeneity among different studies has emerged [63,64]. In a restricted meta-analysis of prospective data, the risk of any fracture was increased by 30% [64]. It has been demonstrated that the increased fracture risk remains up to 20 years after the diagnosis of celiac disease [55,60]. In some studies, the risk of fractures was higher at the wrist [65] or was related to clinical presentation and gender, being higher in symptomatic cases and in males [60]. However, low BMD and bone fragility are also common in asymptomatic patients. In this regard, serologic analyses in cohorts of osteoporotic patients without gastrointestinal manifestations have shown a higher incidence of celiac disease (reaching 9.4% in postmenopausal women with osteoporosis) than in the general population [66,67]. Likewise, in the Tromsø study cohort, serological markers of celiac disease were more frequent in subjects with incidentally detected secondary hyperPTH when compared with matched control subjects [68]. Conversely, a more recent retrospective cohort study in patients visiting a Fracture Liaison Service observed an overall prevalence of serologically and bioptically confirmed celiac disease of 0.38%, which falls within the range of prevalence rates described in the Western-European population (0.33–1.5%) [69].

### 3.2. Pathogenesis of Bone Fragility in Celiac Disease

The pathogenesis of bone damage in celiac disease is multifactorial and includes local and systemic mechanisms [67]. Calcium absorption is generally impaired due to mucosal atrophy, often leading to hypocalcemia; consequently, the parathyroid hormone increases (secondary hyperPTH) and stimulates osteoclast-mediated bone degradation, leading to osteopenia and osteoporosis, and increasing fracture risk [60]. In the presence of persistent hypocalcemia, a decrease in bone mineralization may occur, leading to osteomalacia, or rickets in children. Concomitant hypogonadism can also negatively affect bone quality, further increasing the risk of fractures [60]. At the same time, inflammation and the hypersecretion of pro-inflammatory cytokines (e.g., IL-1, IL-6, and TNF alpha, leading to increased RANKL levels) also contribute to the increase in bone resorption by the osteoclasts [70,71]. In this respect, the neutrophil–to–lymphocyte ratio (NLR), as an inexpensive and widely available marker of inflammation, has been also proposed as a bone loss index in postmenopausal women and as a marker of inflammation in celiac patients [72,73]. Different reports indicate that vitamin D deficiency is common in celiac disease, particularly among children who also showed reduced expression of the vitamin D receptor (VDR) and the two epithelial barrier proteins claudin-2 and E-cadherin, which have important roles in the paracellular pathway and correlate with the histological findings of disease severity [74]. However, reports about the variation in the serum 25-hydroxy vitamin D levels in celiac disease patients at diagnosis and on a gluten-free diet (GFD) remain conflicting [75], likely due to heterogeneity among studies and the complexity of vitamin D metabolism.

The recent development of high-resolution peripheral quantitative computed tomography (HR-pQCT) techniques has offered an opportunity to perform a three-dimensional exploration of the microarchitectural characteristics of bone, information that cannot be derived by dual X-ray absorptiometry (DXA). By this approach, bone microarchitecture impairment has been demonstrated in patients with celiac disease. The trabecular bone was the most affected. Cases with active, newly diagnosed disease showed reduced trabecular density, thinner trabeculae, and a lower trabecular number than unaffected subjects of similar age, with a consequent lower bone strength [76,77].

### 3.3. Management of Bone Fragility in IBD

Several studies have shown that GFD may restore skeletal defects, particularly in children (Figure 2), thus decreasing the risk of fracture in patients with celiac disease [78,79,80,81,82,83]. Generally, from a clinical point of view, after starting GFD, systemic inflammation decreases, the intestinal mucosa heals progressively, and normal gastrointestinal absorption is re-established. Consequently, bone resorption decreases, in part explaining the increase in BMD [83] and, ultimately, the decrease in fracture risk. Moreover, celiac disease patients on GFD undergo a significant decrease in the parathyroid hormone (PTH) levels, together with a significant increase in the calcium and vitamin D concentrations, thus allowing adequate skeletal mineralization. However, the effectiveness of GFD depends on the clinical endpoint addressed, and GFD does not normalize BMD in all patients, even after the recovery of the intestinal mucosa, particularly in older patients [60,81,84,85]. A longitudinal HR-pQCT evaluation in patients with celiac disease demonstrated a greater improvement at the trabecular than at the cortical compartment after 1 year of GFD [86]. While the trabecular BMD and trabecular thickness significantly increased at the distal radius and the tibia, the cortical BMD increased to a lesser extent at the tibia, but did not reach statistical significance at the distal radius, while the cortical thickness decreased significantly at both skeletal sites. Indeed, most HR-pQCT parameters continued to be significantly lower than those of healthy controls of similar age (Figure 3). Of interest, either gender or dietary compliance may also differently affect the skeletal response to GFD in patients with celiac disease [82]. Although with some controversy, more recent studies have suggested a long-term normalization of the rate of fractures in celiac disease patients with strict adherence to both GFD and lifestyle recommendations [82].

Despite the recognized increase in bone fragility in celiac disease, there is no general agreement on which patients should undergo BMD measurement or when the first and subsequent measurements should be performed. Based on the American Gastroenterology Association, patients should be screened by bone densitometry at the time of diagnosis of celiac disease and/or after 1 year of GFD [55]. The latter indication should particularly involve patients with asymptomatic celiac disease. Conversely, DXA measurement was considered unnecessary in pediatric cases with newly diagnosed, asymptomatic disease. Other associations more recently suggested performing BMD screening at diagnosis only in celiac disease adults with classic, symptomatic disease or in those patients with common risk factors for bone fragility and/or high titers of serological markers [87]. A follow-up scan should be performed after 1 year of GFD in osteoporotic/osteopenic cases and after 2 years in the presence of normal BMD levels. Importantly, the increase in the prevalence of osteoporosis among patients with asymptomatic celiac disease provides the rationale for starting GFD for all cases of celiac disease [55]. In addition to BMD, the measurement of serum calcium, 25-hydroxy-vitamin D, and, eventually, PTH should be considered at diagnosis [55]. The use of algorithms, such as the Fracture Risk Assessment Tool (FRAX) [88], might improve the assessment of fracture risk in patients with celiac disease. In fact, when celiac disease is considered as a secondary osteoporosis risk factor or BMD is included in the FRAX assessment, this algorithm may accurately predict fracture risk in CD patients, and it may be therefore used in the clinical setting and management of such patients [89].

Finally, the risk of fracture does not solely depend on increasing BMD and bone microarchitecture, as other risk factors, such as the deterioration of body mass, fat, and muscle compartments, and neuromuscular dysfunction, also contribute to bone fragility in patients with celiac disease. All these factors can be ameliorated by GFD and should be considered and monitored in the management of the skeletal health of patients with celiac disease. Thus, in addition to GFD, all patients should receive education on the importance of lifestyle changes (e.g., avoidance of smoking and excessive alcohol intake, together with engaging in regular weight-bearing exercise) and dietary recommendations, including adequate calcium and vitamin D intake [55]. If necessary, calcium (1000–1500 mg/day) and vitamin D (400–800 IU/day) supplementation should be considered, particularly in those patients deemed at high risk for osteoporosis or with documented osteoporosis [55]. Treatment with bone-active compounds should be restricted to patients with persistence of osteoporosis and/or at high risk of fractures despite 1–2 years of GFD with calcium and vitamin D supplementation [60].

## 4. Gastric Disorders

### 4.1. Helicobacter Pylori Infection

Inflammation and its products (e.g., ILs, TNF-alpha, etc.) are generally considered among the relevant factors affecting bone turnover and potentially causing osteoporosis [90]. In this respect, *Helicobacter pylori* (HP) infection is a common gastric disease caused by an ubiquitary, Gram-negative bacteria that nests in the upper gastrointestinal tract and eludes the host’s immunological response, provoking local inflammation that is considered to be, by the current evidence, the major cofactor in many gastrointestinal diseases, such as acute and chronic gastritis (being responsible for more than 90% of cases of chronic/atrophic gastritis), gastric and duodenal ulcers, gastric mucosa-associated lymphoid tissue lymphoma, and gastric cancer [91]. HP is acquired during childhood via oral–oral and fecal–oral routes and affects more than half of the world’s population, rising to 70% or more in developing countries [91,92,93]. This inflammatory condition may be clinically asymptomatic in up to 70–80% of affected individuals before the occurrence of gastric complications, but it is often associated with gastritis and, unless eradicated by treatment, persists for life [91,92,94]. Moreover, the effects of HP infection may not be confined solely to the digestive tract, but may spread to involve extra-intestinal tissues and/or organs, influencing the clinical course of other common conditions, such as atherosclerosis, ischemic heart disease, insulin resistance, and bone fragility [95,96]. The potential mechanisms underlying these associations remain to be addressed and might go well beyond the enhanced inflammatory state and chronic stimulation of the immune system, e.g., involving increased oxidative stress or autoimmune mechanisms due to molecular mimicry between bacterial peptides and the host [95,96,97,98]. Likewise, the release of different hormonal factors directly from the intestine might be deregulated by chronic HP infection [95,99,100]. Strains carrying virulence factors, such as the cytotoxin-associated gene A (CagA, which is present in 30–40% of patients in Western Countries and in up to 95% in East Asian Countries), are associated with increased production of proinflammatory cytokines and are considered more pathogenic compared to the HP strains lacking these factors [101,102,103].

To date, a limited number of studies have examined the relationship between HP and bone fragility, with conflicting results (Table 1). In a meta-analysis of 21 observational studies with 9655 participants, HP infection was significantly associated with the risk of osteopenia (OR = 1.22, 95% CIs, 1.07–1.39) or osteoporosis (OR = 1.61, 95% CIs, 1.11–2.22) [104]. However, most of the studies were retrospective, cross-sectional, and did not assess anti-CagA antibodies as an indicator of infection with more virulent HP strains [105,106,107,108,109,110,111,112,113,114,115,116,117,118]. Indeed, in a preliminary analysis of a limited sample of elderly men of Caucasian ancestry, we first demonstrated that infection by HP CagA-positive strains was associated with increased cytokine levels and bone resorption markers, and was more prevalent in osteoporotic than non-osteoporotic subjects (OR = 2.13, 95% CIs, 1.02–4.44) [119]. More recently, we replicated this observation in a large, prospective, population-based study involving 1149 adults followed prospectively for up to 11 years [120]. While the overall prevalence of HP infection did not differ among individuals with normal BMD, osteoporosis, and osteopenia, infection by CagA-positive strains was significantly increased in osteoporotic (30%) and osteopenic (26%) patients compared to subjects with normal BMD (21%), which is equivalent to an odds ratio for osteoporosis/osteopenia of 1.34 (95% CIs, 1.01–1.78). Moreover, the anti-CagA antibody levels were significantly and negatively associated with lumbar or femoral BMD and with bone loss (albeit assessed in a subset of cases). Consistent with these associations, CagA-positive cases had a more than five-fold increased risk of sustaining a clinical vertebral fracture (HR = 5.27, 95% CIs = 2.23–12.63) and a double risk to sustaining a non-vertebral incident fracture (HR = 2.09; 95% CIs = 1.27–2.46). Interestingly, reduced estrogen and ghrelin levels, together with an impaired bone turnover balance (in favor of bone resorption, particularly when assessed after the meal) were also observed in carriers of CagA-positive HP infection. A slight increase in the overall occurrence of incident fractures was also evidenced in subjects infected by CagA negative HP strains.

A similar association was reported in a larger retrospective cohort study on the Korean population (*n* = 10,482 women without osteoporosis at baseline), aimed at assessing the risk of incident osteoporosis over 15 years [121]. During the 77,515.3 person-years of follow-up, women with HP infection had a higher rate of incident osteoporosis than those who were uninfected, after adjusting for several potential confounders (HR = 1.23, 95% CIs = 1.03–1.45) [121].

Likewise, two large-scale, retrospective analyses evidenced an increased prevalence of osteoporosis (as assessed by bone densitometry criteria) in HP-positive patients of Asian ancestry than in unaffected controls [105,122]. In particular, an increased risk of developing osteoporosis was reported among more than 5000 patients who received HP eradication therapy than a reference cohort of age and sex-matched unaffected controls [105]. The late eradication group, which had a longer period of chronic HP infection, had a higher incidence of osteoporosis than the early eradication group. Moreover, the risk remained statistically significant for up to 5 years after eradication, thus suggesting a major and long-lasting negative effect of HP on the skeleton. Since most HP strains infecting people of Asian descent have been reported to express CagA [101,123], the results of these studies (both performed in Taiwan) are consistent with our findings and similar to previous observations in other conditions, such as coronary heart disease or diabetes, suggesting an association between these disorders and chronic CagA-positive HP infection [124,125,126,127]. Thus, the genetic heterogeneity of strains may also explain some of the discordant findings about the association between HP and osteoporosis or other chronic disorders.

The mechanisms by which HP infection causes bone fragility and increases fracture risk remain to be elucidated in detail and might involve systemic inflammation, impaired gastric acidification, impairment of endocrine function, increased oxidative stress, or autoimmunity (Figure 4). Indeed, there is increasing evidence that an excessive inflammation response, in addition to dysregulated hormonal status, plays an important role in the pathogenesis of bone loss and, consequently, of osteoporosis in both genders [92,128]. Chronic HP infection induces an activation of most components of innate and adaptive immunity, causing a prolonged activation of local and systemic inflammatory responses that lead to atrophic gastritis and malabsorption. This also includes increased production of major pro-osteoclastogenic cytokines, such as TNFα, IL-1, IL-6, and IL-8, that enhance bone resorption and bone loss. At the same time, atrophic gastritis and/or the use of antisecretory drugs affecting acid secretion might lead to the malabsorption of dietary and supplementary calcium, thereby increasing the risk of osteoporosis over the long term [129,130,131]. The skeletal implications of HP-induced inflammation could become particularly relevant with aging due to the decrease in estrogen levels that can be observed in both genders, and particularly in women after menopause [132].

Among the hormonal factors, a reduction in circulating ghrelin levels has been described in HP infection, caused by the destruction of the oxyntic glands of the stomach [120,133,134]. Together with its effects on the stimulation of food intake, ghrelin directly affects osteoblasts, stimulating their proliferation and differentiation [135]. Therefore, a reduction in the circulating ghrelin levels could decrease bone formation and negatively impact bone health. Moreover, HP infection has been associated with a reduction in the total, free, and bioavailable estradiol levels in both genders [120], likely due to a HP-induced loss of gastric parietal cells expressing aromatase and thus able to convert circulating androgen precursors into estrogen [136,137]. This impairment appears particularly relevant in the presence of CagA-positive strains. Thus, consistent with preclinical observations [136,138], it is likely that, in elderly subjects, the peripheral aromatization of androgen precursors into estrogen by parietal cells of the gastric mucosa may, in part, contribute to the overall circulating pool of estradiol in the elderly, with potential implications on skeletal health [120]. Finally, HP infection, primarily by CagA-positive strains (due to extended damage of the gastro–duodenal mucosa), may also affect the release of gastrointestinal peptides, with additional negative effects on bone turnover [120,139].

No specific indications have been released to date concerning the management of bone fragility and the prevention of fractures in subjects affected with HP infection.

### 4.2. Chronic Gastritis and Peptic Ulcer Disease

The skeletal implications of hypochlorhydria have received much attention, as several observational studies and meta-analyses showed an association between the use of proton pump inhibitors (PPIs) and increased fracture risk, mainly hip fractures [140,141,142]. However, in these studies, the effects of different confounding variables that could potentially affect bone health in PPIs users were not adequately considered [142]. Thus, although this association remains debated, a more pronounced reduction in gastric acid secretion is usually seen in patients with chronic atrophic gastritis, thus explaining the reported association between this condition, osteoporosis, and fracture risk, described in some, but not all, studies [109,114,131,143,144,145], particularly in the presence of autoimmune atrophic gastritis and pernicious anemia [146,147]. In fact, many patients express autoantibodies against parietal cells and intrinsic factors leading to the malabsorption of vitamin B12 and pernicious anemia, with possible negative skeletal effects. In addition to direct skeletal effects, vitamin B12 depletion induces peripheral neuropathy [148], thereby increasing the risk of falls. Impaired bone quality (as indirectly suggested by decreased trabecular bone score), in addition to reduced BMD, has also been described in patients with chronic atrophic gastritis [143]. Moreover, hypochlorhydria, and particularly anachlorhydria, may negatively affect calcium absorption and impair normal bone mineralization, since mice deficient in *Cckbr* (encoding a gastrin receptor that affects acid secretion by parietal cells) have hypocalcemia, secondary hyperPTH, and osteoporosis [129].

Since HP is responsible for more than 90% of chronic atrophic gastritis and plays an important role in the pathogenesis of autoimmune atrophic gastritis (due to molecular mimicry between HP antigens and gastric H/K-ATPase), other additional HP-related mechanisms (as described in the previous chapter) might explain bone fragility in patients with atrophic gastritis. This should also explain the association between peptic ulcer disease and osteoporosis reported by several studies [149,150,151]. In fact, HP is responsible for 85% of gastric ulcers and 95% of duodenal ulcers [152]. In particular, an increased risk of osteoporosis was described in two large studies from Korea (with HR ranging from 1.36 to 1.72) [149,150], as well as in a population-based study from Taiwan (HR = 1.85, 95% CIs = 1.73–1.98) [153]. In the latter study, a particularly higher risk was observed in the first year after peptic ulcer diagnosis (HR = 63.4, 95% CIs = 28.2–142.7). Importantly, the incidence of osteoporosis was consistently higher in the peptic ulcer group at all ages, but the age-specific risk analysis showed that the risk of osteoporosis was higher in patients under 50 years (HR = 6.15, 95% CIs = 4.68–8.09) than in older cases (HR = 1.71, 95% CIs = 1.60–1.83, *p* < 0.001). Consistent with those observations, an increased risk of periprosthetic fractures was also observed in North American cohorts of patients with peptic ulcer disease who underwent total knee or total hip arthroplasty [154,155].

### 4.3. Gastric Cancer and Gastrectomy

Gastric cancer is the fifth-most common cancer worldwide and remains amongst the more frequent causes of cancer-related mortality. However, in recent times, there has been a considerable improvement in the early diagnosis and management of this disorder (with a combination of radical surgery and adjuvant therapy), so that survival rates have markedly improved (with a 5-year survival rate of around 80–90% in most countries) [156]. With the increasing number of long-term survivors after gastrectomy, there has been increasing interest in the clinical management of post-gastrectomy syndromes and their possible complications. Among these complications, several reports indicated that patients with gastric cancer, and particularly gastric cancer survivors who underwent gastrectomy, have an increased risk of osteoporosis and fragility fractures [153,157,158,159,160,161,162,163,164,165,166,167,168]. This is in keeping with previous observations in patients with gastrectomy for peptic ulcer disease [169]. In particular, prevalence rates of around 38–55% and 40% for osteoporosis and fractures, respectively, were described in gastric cancer survivors who underwent gastrectomy (revised by [152]). It seems that the highest risk is observed in the early post-operative period, within 1 to 5 years post-gastrectomy [160,163], but may persist at higher rates than in the general population for up to 20 years following surgery [166]. However, much of the evidence comes from cross-sectional or retrospective reports, and the indications from the few prospective studies are somewhat divergent. While a first prospective analysis in a small sample of men with early gastric cancer who underwent gastrectomy showed that BMD loss mainly occurs within the first post-operative year (−4% in the first year and −1% in the second year after surgery) [165], in a more recent study of gastrectomized men, the increase in bone turnover and the loss of BMD persisted for up to 20 years after surgery [166]. In the latter cohort, after adjusting for confounding factors, history of gastrectomy was associated with an increased risk of incident osteoporotic fractures (HR = 2.81, 95% CIs = 1.28–6.18) that was highest in men who survived 20 years or more after the surgery (HR = 3.56, 95% CIs = 1.33–9.52). However, when classified by reason for gastrectomy, those who underwent gastrectomy for peptic ulcers showed a significantly increased risk of fracture (HR = 3.71, 95% CIs = 1.43–9.60), but those who underwent gastrectomy for cancer did not (HR = 2.00, 95% CIs = 0.59–6.64).

Despite some studies suggesting an increased risk of osteoporosis after total gastrectomy than partial gastrectomy [170], or Roux-en-Y than Billroth-I reconstruction [171], it remains unclear which surgical procedure for gastric cancer confers a greater risk to bone health. Indeed, a certain degree of bone loss has also been described in a restricted cohort of gastric cancer survivors with early stage gastric cancer after endoscopic tumor resection [162]. In addition to osteoporosis and fractures, calcium malabsorption, together with an increased risk of osteomalacia, has also been described after gastrectomy [157,172]. In particular, the incidence of osteomalacia after gastrectomy has been approximately estimated to be around 10–20% [55].

The pathogenesis of bone fragility in gastric cancer survivors after gastrectomy remains to be, in part, understood and is probably related to different mechanisms, such as malabsorption of calcium and vitamin D (leading to secondary hyperPTH), and protein malnutrition. Indeed, gastrointestinal physiology could be altered after gastrectomy and reconstruction, particularly in the duodenum and proximal jejunum, which are the main sites of calcium absorption. The presence of hyperPTH may explain the imbalance in bone turnover often described post-gastrectomy, with an increase in bone resorption markers but without concomitant, persistent increases in bone formation markers. The decreased ability to absorb lipids results in reduced absorption of all liposoluble vitamins in addition to vitamin D [173], such as vitamin K, which is also implicated in skeletal health [174]. Although this issue remains debated and might involve a subset of gastrectomized patients, a detailed study assessing different vitamin D metabolites reported the presence of vitamin D deficiency in 32% of patients [175]. Moreover, during the immediate postoperative period, most patients may experience rapid weight loss (ranging from 5% to 15%), which is likely related not only to malabsorption, but also to a lack of appetite, dyspepsia, and altered intestinal motility [172,176,177,178]. The loss of body weight might directly and indirectly impact bone tissue by increasing bone resorption due to reduced mechanical loading or causing a dysregulation of hormonal pathways (e.g., ghrelin, leptin, and adiponectin) [179]. Thus, a low BMI before gastrectomy has been proposed as a reliable predictor of osteoporosis and bone fragility after gastric cancer surgery [153]. Likewise, the presence of complications, such as diabetes or dyslipidemia, as well as chemotherapy, have been associated with a higher risk of fracture after gastrectomy [163]. Anemia and a reduction in hemoglobin levels have been also linked to a higher risk of bone fragility in gastric cancer patients [167,168]. Although the underlying mechanism remains to be fully understood, it is likely that increased oxidative stress and extracellular acidification under hypoxic conditions might negatively affect bone remodeling, as also observed in the general population [180,181].

The management of bone fragility after gastrectomy has only been addressed in a 2003 report from the American Gastroenterological Association, recommending a BMD screening in patients who are at least 10 years post-gastrectomy, particularly in case of postmenopausal females and males over 50 years of age, or in the presence of fragility fracture history [55]. These recommendations were mostly derived from reports involving patients with peptic ulcer disease, and might differ in patients with gastric cancer due to their older age, their worse general conditions, and the observed increase in fracture incidence in the short term after surgery. Thus, it is likely that specific and earlier recommendations are required in this setting. Finally, although the optimal therapeutic strategy for this condition remains to be established, some reports suggested that vitamin D or its active metabolites (e.g., alfacalcidol) might at least maintain the BMD values and prevent gastrectomy-induced bone loss [182,183]. In a single, small-scale, preliminary study, the superiority of alendronate plus alfacalcidol treatment over alfacalcidol in improving BMD was demonstrated [184].

## 5. Microbiome, Dysbiosis, and Bone Health

Over the past few years, different reports indicated that the loss of intestinal barrier function by either food-derived environmental triggers or changes in the composition of the array of microorganisms residing within the human intestine (the so-called gut microbiota) might affect the immune homeostasis and might be involved in the pathogenesis of inflammation and common metabolic disorders, including osteoporosis [185,186,187]. The intestinal microbiota comprise a highly diverse population of more than 10^13^–10^14^ bacteria, representing 5000 species and 5 million genes (the so-called microbiome) [188]. These microorganisms have been shown to elicit either positive or negative influences on host health by the regulation of vitamin production, nutrient, and energy extraction from the diet, metabolic function, regulation of barrier integrity, and modulation of the local and systemic immune system [187,189,190].

To date, the potential contribution of the intestinal microbiome to skeletal homeostasis has not yet been fully established [191,192,193]. However, different indirect evidence has demonstrated positive effects of either prebiotics or probiotics on bone [191]. Prebiotics are non-digestible nutrients, such as fibers and oligosaccharides, which can modulate the intestinal bacterial communities with positive effects on the host. The most-studied for their effect on bone are inulin, oligo-fructose, and galacto-oligosaccharides. These compounds may positively impact calcium homeostasis because they are transformed by the fermentation processes of the microbiota into fatty acids, which promote calcium solubilization and its absorption by acidifying the intestinal lumen. Importantly, fatty acids, such as butyrate, have also been identified as positive mediators of bone formation [194,195]. On the other hand, probiotics are microorganisms that confer beneficial effects to the health of the host when ingested. Initial studies on animal models demonstrated that probiotic use (e.g., *Lactobacillus reuteri*) decreases intestinal inflammation, stimulates osteoblastic activity, reduces osteoclastic bone resorption (through the inhibition of TNF-alpha production), and increases bone density in male mice [196]. Likewise, in female mice, probiotics were able to prevent ovariectomy-induced bone loss by reducing osteoclast activation, and this effect was associated with a modification of the bacterial diversity in the intestine [197,198,199].

More recent and complex animal studies suggested that estrogen and the gastrointestinal microbiota might synergize to influence intestinal permeability and different metabolic pathways [200]. In particular, steroid hormone deficiency may induce an increase in intestinal permeability and an up-regulation of some cytokines (e.g., TNF-alpha, RANKL, IL-17) in the small intestine and bone marrow, which can induce osteoclast activation and bone loss [201]. This effect was absent in germ-free mice models that are resistant to bone loss induced by estrogen depletion, as well as in estrogen-depleted mice after the use of probiotics, such as *Lactobacillus rhamnosus* or VSL#3 (a formulation of different probiotic strains). Thus, the intestinal microbiome, together with its influence on gastrointestinal permeability, plays a relevant role in triggering the inflammatory pathways that are critical for osteoclast activation during estrogen deficiency, suggesting that the use of probiotics might be beneficial as a therapeutic approach for postmenopausal osteoporosis. Consistent with this hypothesis, short-term trials performed in postmenopausal women suggested that the use of probiotics might attenuate bone loss [202,203,204,205,206]. Large-scale prospective trials are required to specifically evaluate the skeletal response to different probiotics and establish whether the positive effect on bone are sustained over the longer term. Likewise, changes in the microbiome composition between osteoporotic and non-osteoporotic subjects have recently been suggested by preliminary observational reports in small cohorts (e.g., reporting an increase in certain species, such as Bacteroides, and decreases in Firmicutes, Actinobacteria, or unclassified Clostridia in osteoporosis) [207,208,209], which require to be replicated in large-scale prospective studies. Taken together, these observations suggest that intestinal microbiota dysbiosis, in addition to specific gastrointestinal diseases (and their potential implications on the microbiota), might play a relevant role in bone fragility.

## 6. Conclusions

Osteoporosis and fragility fractures are very common conditions and are increasingly recognized as a source of significant morbidity and disability, leading to a huge health and economic burden in most countries [1,2]. Despite their wide prevalence in the general population, the skeletal implications of many gastrointestinal diseases have been poorly investigated, and their potential contribution to bone fragility is often underestimated in clinical practice. Indeed, proper functioning of the gastrointestinal system appears essential for the skeleton, allowing correct absorption of calcium, vitamins, or other nutrients, preserving the gastrointestinal barrier function, and maintaining an optimal endocrine-metabolic balance, so that it is very likely that most chronic diseases of the gastrointestinal tract, and even gastrointestinal dysbiosis, may have profound implications for bone health. While the association between celiac disease or IBD and increased risk of osteoporosis is generally recognized [55,210], the negative skeletal implications of common diseases affecting the stomach, such as gastritis, peptic ulcers, and gastric cancer, have been poorly investigated. However, recent evidence from either prospective or observational studies (also including retrospective observations in large population samples) suggests that the presence of HP infection (particularly in the presence of the CagA virulent strain), which is the leading cause of gastric disorders, is associated with increased risk of osteoporosis and fragility fractures. Likewise, different reports suggest that post-gastrectomy syndromes (either due to peptic ulcer disease or gastric cancer) may negatively affect bone health [55,210]. An updated summary with a tentative list of diagnostic and therapeutic indications for the management of bone health in patients with major gastrointestinal disorders is given in Table 2 At the same time, there is very limited information arising from randomized controlled trials concerning the effects of preventive and therapeutic interventions with antiresorptive or bone-anabolic agents on bone health in patients with gastrointestinal disorders [210]. In fact, most of the information about the management of fracture risk in these patients derives from studies in postmenopausal women or in subjects with glucocorticoid-induced osteoporosis [46,211]. We therefore hope that future studies might address these gaps so that more specific indications will be released in the future for adequate management of bone health in patients with gastrointestinal pathologies.

## Figures and Tables

**Figure 1 ijms-23-02713-f001:**
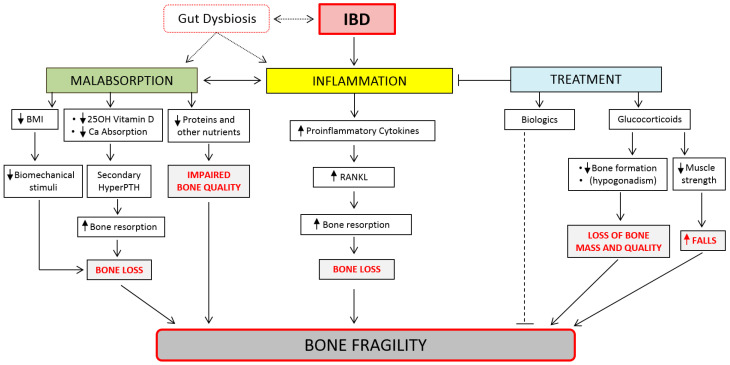
Pathophysiology bone fragility induced by Inflammatory Bowel Disease (IBD). (HyperPTH: hyperparathyroidism).

**Figure 2 ijms-23-02713-f002:**
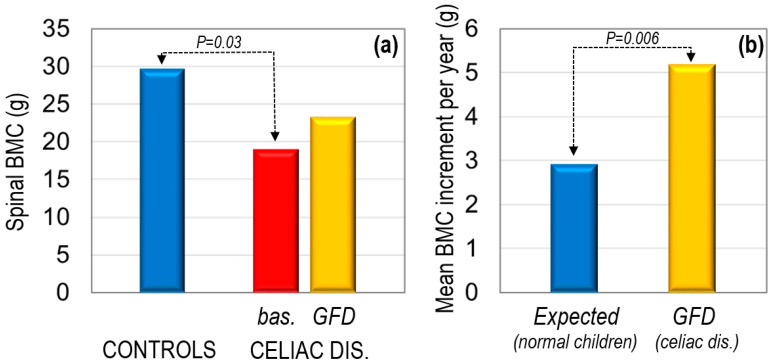
Effects of a gluten-free diet (GFD) on the spinal bone mineral content (BMC) (**a**) as assessed by the dual X-ray absorptiometry (DXA) technique and annual BMC increment and (**b**) in children with celiac disease (*n* = 22) as compared with 428 healthy, age-matched controls (derived from Barera G et al. [81]). Bas: baseline.

**Figure 3 ijms-23-02713-f003:**
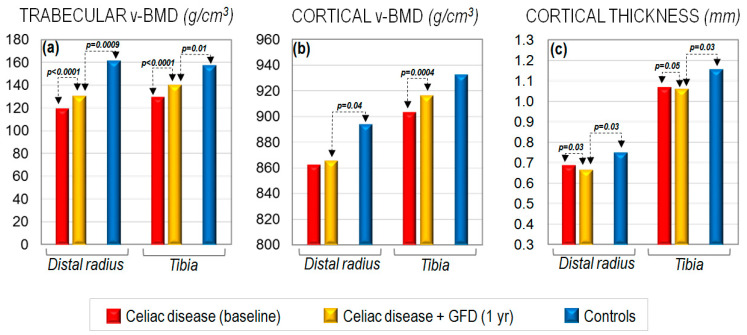
Trabecular (**a**) and cortical (**b**) volumetric BMD changes at the distal radius or tibia and variation of cortical thickness (**c**) assessed with high-resolution peripheral quantitative computed tomography (HRpQCT), before and after 1 year on a gluten-free diet (GFD) in a cohort of premenopausal women with newly diagnosed celiac disease (CD), as compared with healthy females (CTs) of similar age and BMI (derived from Zanchetta MB et al. [86]).

**Figure 4 ijms-23-02713-f004:**
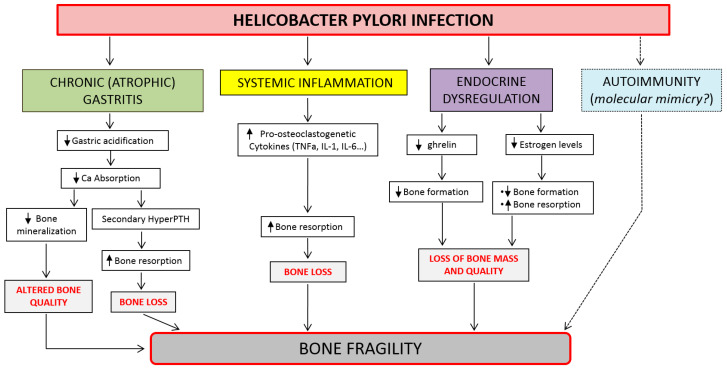
Pathophysiology of bone fragility induced by *Helicobacter Pylori* infection.

**Table 1 ijms-23-02713-t001:** Major studies assessing the relationship between *Helicobacter Pylori* (HP) infection and osteoporosis or fractures.

Reference	Country	Study Design	Sample	HP Cases	Detection Method	Summary of Results
Figura et al., 2005 [119]	Italy	CS	240 men (*55–82 years*)	80	ELISA	Increased OP prevalence in CagA-HP pts (OR 2.1; 95% CIs, 1.0–4.4)
Kakehashi et al., 2009 [114]	Brazil	CS	85 postmen. women	34	UBT/His	No differences in BMD
Akkaya et al., 2011 [107]	Turkey	CS	108 postmen. women	76	ELISA	No association with OP diagnosis
Asaoka et 2014 [115]	Japan	CS	200 men and women (>*50 years*)	83	UBT/ELISA	Increased HP prevalence in OP pts (OR 5.3; 95% CIs, 1.7–16.4)
Lin et 2014 [108]	Taiwan	CS, R	365 women (>*65 years*)	77	UBT/His	Increased HP prevalence in OP pts (OR 2.0; 95% CIs, 1.1–3.6)
Fotouk-Kiai et al., 2015 [111]	Iran	CS	967 men and women (>*60 years*)	758	ELISA	No differences in BMD or OP prevalence (OR 0.76; 95% CIs, 0.6–1.0)
Chung et al., 2015 [112]	Korea	CS	1126 men	469	ELISA	Decreased LS-BMD in HP pts
Asaoka et al., 2015 [110]	Japan	CS	255 men and women (>*50 years*)	94	UBT/ELISA	Increased HP prevalence in OP pts (OR 3.0; 95% CIs, 1.3–6.9)
Mizuno et al., 2015 [109]	Japan	CS	230 men (*50–60 years*)	99	ELISA	Increased HP prevalence in OP pts (OR 1.8; 95% CIs, 1.0–3.2)
Kalantarhormozi et al., 2016 [113]	Iran	P	250 postmen. women (*55–82 years*)	143	ELISA	No difference in OP incidence (OR 0.96; 95% CIs, 0.5–1.8)
Chinda et al., 2017 [116]	Japan	CS	473 women (*20–86 years*)	118	ELISA	No difference in osteopenia prevalence (OR 0.95; 95% CIs, 0.5–1.6)
Lu et al., 2018 [117]	China	CS	1867 men and women	589	UBT	No association with OP diagnosis
Pan et al., 2018 [105]	Taiwan	CS	867 men and women (>*20 years*)	381	His	Increased HP prevalence in OP pts (OR 1.6; 95% CIs, 1.1–2.3)
Chinda et al., 2019 [118]	Japan	CS, R	268 men (*19–90 years*)	268	ELISA	No association with the diagnosis of osteopenia
Gennari et al., 2021 [120]	Italy	P	1149 men and women (*50–80 years*)	566	ELISA	Increased fracture incidence in HP pts (HR 1.9; 95% CIs, 1.1–2.9) Increased fracture incidence in CagA-HP pts (HR 2.0; 95% CIs, 1.2–2.9)
Kim et al., 2021 [121]	Korea	R	10482 women (>*20 years*)	6009	ELISA	Increased OP incidence in HP pts (HR 1.2; 95% CIs, 1.0–1.4)

CS = cross-sectional; R = retrospective, P = prospective. ELISA = ELISA anti-HP IgG assays; UBT = carbon-13 urea breath test; His = histology. OP = osteoporosis (WHO criteria); BMD = bone mineral density; HP = Helicobacter Pylori; CagA-HP = CagA-positive Helicobacter Pylori; pts = patients.

**Table 2 ijms-23-02713-t002:** Suggested diagnostic and therapeutic indications for the management of bone health in patients with major gastrointestinal disorders.

Disease	Screening Indications	Follow-Up Indications	Treatment Indications *
IBD	DXA at diagnosis if: *Postmenopausal women or men > 50 years* *Previous fragility fractures* *GC treatment or other risk factors for osteoporosis*	Repeat DXA within 2 years if: *Low BMD at baseline (T score < −1.0)* *Long-term GC treatment*	Consider treatment with bone-active agents if: *Osteoporosis (BMD T score < −2.5)* *Fragility fracture* *Long-term GC treatment*
Celiac Disease	DXA at diagnosis if *Adult patients (>30 years)* *Previous fragility fractures* *≥1 risk factors for osteoporosis*	Repeat DXA within 1–2 years after GFD if: *Low BMD at baseline (T score < −1.0)* *Postmenopausal women or men > 50 years*	Consider treatment with bone-active agents if: *Osteoporosis (BMD T score < −2.5)* *Fragility fracture*
Helicobacter Pylori Infection	DXA at diagnosis if: *Postmenopausal women or men > 50 years* *CagA-positive strain* *Previous fragility fractures*	Repeat DXA within 1–2 years after eradication if: *Low BMD at baseline (T score < −1.0)* *Postmenopausal women or men > 50 years*	Consider treatment with bone-active agents if: *Osteoporosis (BMD T score < −2.5)* *Fragility fracture*
Chronic Gastritis or Peptic Ulcer Disease	DXA at diagnosis if: *Postmenopausal women or men > 50 years* *Previous fragility fractures* *Pernicious anemia* *CagA-positive Helicobacter Pylori infection*	Repeat DXA within 2 years if: *Low BMD at baseline (T score < −1.0)*	Consider treatment with bone-active agents if: *Osteoporosis (BMD T score < −2.5)* *Fragility fracture*
Post-gastrectomy	DXA within 1–2 years post-gastrectomy	Repeat DXA within 2 years if: *Low BMD at baseline (T score < −1.0)* *Persistent weight loss (>5%)*	Consider treatment with bone active agents if: *Osteoporosis (BMD T score < −2.5)* *Fragility fracture*

* Education on the importance of lifestyle changes (e.g., regular exercise, smoking cessation, and avoidance of alcohol abuse) and adequate calcium intake should be recommended in all patients. Vitamin D deficiency (25-hydroxyvitamin D levels < 20 ng/mL) should be also identified and corrected. DXA = dual X-ray absorptiometry; GC = glucocorticoid; BMD = bone mineral density.

## Data Availability

Not applicable.

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
