# Peer review of "Bone Fragility in Gastrointestinal Disorders"

_ijms, 2022, doi:10.3390/ijms23052713_

Round 1

Reviewer 1 Report

This is generally well written review. I have just a few minor editorial comments.

Throughout the whole manuscript – please be consistent – CI or CIs, Odds Ratio or OR,  HR or hazard ratio, with or without any symbols like “,” , “:” or “=” )

Please also correct helicobacter pylori to Helicobacter pylori (L396, Table 1, Figure 4 and others) and write it in italic (the later remark applies to all other names in L648, L660, L671)

L37-40 reference needed

L55 Please consider introduction here the term "secondary osteoporosis" which is the main topic of this work and special issue. In fact, the term "secondary osteoporosis"  appears only once in your work (L374)

L89 repetition of L68

L95 (anti-TNF)

L106 T-score

L145 factor-kappa B

L182 Please introduce the HyperPTH abbreviation here which is present in Fig 1.

L249 Randomized controlled trials

L253-255 reference needed

L311 claudin-2 (lowercase)

L326 GFD was introduced in L315 as gluten-free

L332 PTH abbreviation

L317 “25-hydroxyvitamin D3 (calcifediol)”

L397 Gram-negative

L519 B12

L630 correct to „100 trillion” or 10^14

Thank you for the opportunity of reviewing this manuscript.

Author Response

We thank the reviewer for his valuable comments. We reply point by point to his questions. In detail:

  • Throughout the whole manuscript – please be consistent – CI or CIs, Odds Ratio or OR,  HR or hazard ratio, with or without any symbols like “,” , “:” or “=” )

We indicate always “Odds ratio” as “OR”, “CIs=” as intervals of confidence and “HR=” as hazard ratio throughout the whole manuscript

  • Please also correct helicobacter pylori to Helicobacter pylori (L396, Table 1, Figure 4 and others) and write it in italic (the later remark applies to all other names in L648, L660, L671)

The term Helicobacter pylori has been corrected throughout the whole manuscript and written in italic.

  • L37-40 reference needed

References have been added

  • L55 Please consider introduction here the term "secondary osteoporosis" which is the main topic of this work and special issue. In fact, the term "secondary osteoporosis"  appears only once in your work (L374)

As suggested, the sentence in line 55 has been revised, giving directly the term “secondary osteoporosis”.

  • L89 repetition of L68

The sentence beginning at line 89 has been removed

  • L95 (anti-TNF)

OK, it has been corrected

  • L106 T-score

OK, it has been corrected

  • L145 factor-kappa B

OK, it has been corrected

  • L182 Please introduce the HyperPTH abbreviation here which is present in Fig 1.

The abbreviation “HyperPTH” for hyperparathyroidism has been directly specified in the legend of Figure 1 rather than in the text.

  • L249 Randomized controlled trials

OK, it has been corrected

  • L253-255 reference needed

References have been added

  • L311 claudin-2 (lowercase)

OK, it has been corrected

  • L326 GFD was introduced in L315 as gluten-free

OK, it has been corrected

  • L332 PTH abbreviation

OK, it has been corrected

  • L317 “25-hydroxyvitamin D3 (calcifediol)”

OK, it has been corrected

  • L397 Gram-negative

 OK, it has been corrected

  • L519 B12

 OK, it has been corrected

  • L630 correct to „100 trillion” or 10^14.

The sentence has been corrected in “The intestinal microbiota comprises a highly diverse population of more than 1013–1014 bacteria, representing 5000 species and 5 million of genes”.

Reviewer 2 Report

I consider that the manuscript entitled “Bone Fragility in Gastrointestinal Disorders” is well written and provides valuable information in the field. It provides an elaborated review of the influence of gastrointestinal disorders and bone metabolism, therefore I congratulate the authors for their intensive work. I recommend for publication in the International Journal of Molecular Sciences after minor revision.                   

  • Please add information on the proton pump inhibitor action on bone metabolism. They are often used in the gastrointestinal disorders and the information in the manuscript is limited.
  • Please insert a table with the recommended screening program and recommended treatment for each of the gastrointestinal disorders. I consider that it will provide benefits for clinicians.
  • Replace “gut” with ”intestinal” or “gastrointestinal”
  • Line 37: Replace “skeletal districts”with “other locations”
  • Figure 1: why is hypognonadism in brackets?
  • Line 231: “50 years”
  • Figure 2: please explain “bas”
  • Line 394: “(e.g. interleukins, tumor necrosis factor-alpha)”
  • Line 422: meta-analysis
  • Line 536: “[158]. In the latter …”
  • Line 680: “…in most countries”

Author Response

We thank the reviewer for his valuable comments. We reply point by point to his questions. In detail:

  • Please add information on the proton pump inhibitor action on bone metabolism. They are often used in the gastrointestinal disorders and the information in the manuscript is limited.

We agree about the relevance of proton pump inhibitors for bone disorders. However the manuscript was focused of gastrointestinal diseases and osteoporosis. The addition of information about the skeletal implications of this drug category would require ample space if not a separate chapter and is outside the scope of this paper. Following the reviewer suggestion, we include a statement with a new reference about a very recent paper revising the role of proton pump inhibitors in skeletal health.

  • Please insert a table with the recommended screening program and recommended treatment for each of the gastrointestinal disorders. I consider that it will provide benefits for clinicians.

We thank the Reviewer for the suggestion. We have included a new Table (Table 2) reporting a tentative list of diagnostic and therapeutic indications for the management of bone health in patients with major gastrointestinal disorders.

  • Replace “gut” with ”intestinal” or “gastrointestinal

The term “gut” has been replaced throughout the whole manuscript with ”intestinal” or “gastrointestinal”.

  • Line 37: Replace “skeletal districts”with “other locations

As suggested, the term “districts” has been replaced with skeletal sites

  • Figure 1: why is hypognonadism in brackets?

It is in parenthesis and not in brackets because does not involve most patients on glucocorticoid treatment (in contrast with decreased bone formation that is generally common under long term glucocorticoid treatment), but it is often reported as additional risk factor to explain bone fragility in glucocorticoid users.

  • Line 231: “50 years”

OK, it has been corrected

  • Figure 2: please explain “bas”

The term “bas” indicates “ baseline” and it has been explained in the legend of the Figure 2

  • Line 394: “(e.g. interleukins, tumor necrosis factor-alpha)”

OK, it has been corrected

  • Line 422: meta-analysis

OK, it has been corrected

  • Line 536: “[158]. In the latter …”

OK, it has been corrected

  • Line 680: “…in most countries”

OK, it has been corrected
